# Rapid geographical source attribution of *Salmonella enterica* serovar Enteritidis genomes using hierarchical machine learning

Sion C Bayliss[1]*, Rebecca K Locke[2,3], Claire Jenkins[4], Marie Anne Chattaway[4], Timothy J Dallman[5], Lauren A Cowley[2]

[1]Bristol Veterinary School, University of Bristol, Bristol, United Kingdom; [2]Milner Centre for Evolution, Life Sciences Department, University of Bath, Bath, United Kingdom; [3]Genomic Laboratory Hub (GLH), Addenbrooke's Hospital, Cambridge University Hospitals NHS Foundation Trust, Cambridge, United Kingdom; [4]Gastrointestinal Reference Services, UK Health Security Agency, London, United Kingdom; [5]Institute for Risk Assessment Sciences, Utrecht University, Utrecht, Netherlands

*For correspondence:
s.bayliss@bristol.ac.uk

**Competing interest:** The authors declare that no competing interests exist.

**Abstract** *Salmonella enterica* serovar Enteritidis is one of the most frequent causes of Salmonellosis globally and is commonly transmitted from animals to humans by the consumption of contaminated foodstuffs. In the UK and many other countries in the Global North, a significant proportion of cases are caused by the consumption of imported food products or contracted during foreign travel, therefore, making the rapid identification of the geographical source of new infections a requirement for robust public health outbreak investigations. Herein, we detail the development and application of a hierarchical machine learning model to rapidly identify and trace the geographical source of *S.* Enteritidis infections from whole genome sequencing data. 2313 *S.* Enteritidis genomes, collected by the UKHSA between 2014–2019, were used to train a 'local classifier per node' hierarchical classifier to attribute isolates to four continents, 11 sub-regions, and 38 countries (53 classes). The highest classification accuracy was achieved at the continental level followed by the sub-regional and country levels (macro F1: 0.954, 0.718, 0.661, respectively). A number of countries commonly visited by UK travelers were predicted with high accuracy (hF1: >0.9). Longitudinal analysis and validation with publicly accessible international samples indicated that predictions were robust to prospective external datasets. The hierarchical machine learning framework provided granular geographical source prediction directly from sequencing reads in <4 min per sample, facilitating rapid outbreak resolution and real-time genomic epidemiology. The results suggest additional application to a broader range of pathogens and other geographically structured problems, such as antimicrobial resistance prediction, is warranted.

## Editor's evaluation

This important study presents a machine learning-based classifier that can accurately determine the geographic origin of a *Salmonella enterica* sample from its whole-genome sequencing data in under five minutes leading to actionable public health insights. Applying the method to 2,313 whole genome sequences collected in the United Kingdom and several external validation datasets, the authors provide convincing evidence that *Salmonella* genomic data can be used to identify the likely geographic source of a food-borne outbreak and, in most cases, correctly identify the country of

origin of an infection acquired overseas. The work presents an excellent case for the potential utility of routine genomics coupled with machine learning for public health microbiology and the methods are likely to be applicable to other pathogens besides *Salmonella enterica*.

## Introduction

Diarrhoeal diseases are the most common illnesses caused by contaminated food, with 550 million people falling ill each year, including 220 million children under the age of 5 years (**WHO, 2022**). High disease-burden foodborne pathogens represent a major public health concern, necessitating real-time epidemiological monitoring and follow-up. Outbreak investigations are often confounded by the complexity of the international food-trade networks which distribute zoonotic food-borne pathogens across the globe (**Gould et al., 2017**). Understanding the contributing factors, whether they be environmental, geographical, or zoonotic is critical for designing public health intervention strategies to combat and prevent food-borne pathogen outbreaks (**Pires et al., 2009**).

In many developed nations, *Salmonella enterica* serovar Enteritidis is the most common cause of the diarrhoeal disease (**UKHSA, 2021**) and represents a significant economic and healthcare burden (**Daniel et al., 2020**). In the UK, nationwide vaccination and monitoring programs have been responsible for a precipitous drop in detectable *Salmonella* from local animal products and a concurrent drop in human infection rates (**Surveillance, Zoonoses, Epidemiology and Risk Food and Farming Group, 2007**; **Tam et al., 2012**). However, recent studies have identified imported foodstuffs (**Cowley et al., 2016**; **McLauchlin et al., 2019**; **Somorin et al., 2021**) and foreign travel (**PHE, 2017**) as more pertinent *Salmonella* infection risks.

Whole genome sequencing (WGS) has become a powerful tool for untangling complex networks of pathogen dissemination by providing high-resolution sub-typing for transmission tracing as well as detailed information on antimicrobial and virulence status (**Allard et al., 2018**). Since 2014, the UK Health Security Agency (UKHSA) has routinely applied WGS to all clinically identified cases of *Salmonella* in the UK alongside collecting detailed metadata, such as patients recent foreign travel (**Ashton et al., 2016**). This program has been instrumental in identifying various international outbreaks, such as the largest known salmonellosis outbreak in Europe where, between 2015 and 2018 *S*. Enteritidis-contaminated eggs resulted in 1209 reported cases across 16 countries (**Dallman et al., 2016**; **Pijnacker et al., 2019**). The approach taken by the UKHSA for inferring geographical source from genomics data involves phylogenetic population structure analysis, the examination of the genetic relatedness of isolates from various relevant source populations, alongside in-depth analysis of isolates from recent-reported travel cases and reconstruction of domestic or international food networks (**Chan et al., 2023**; **Chattaway et al., 2023**; **Jenkins et al., 2019**). Due to the resources involved, this type of investigation is often only undertaken in exceptional cases representing a threat to public health. A range of other methodologies exist for application to source attribution of genomics data (**Mughini-Gras et al., 2018**) including population genetics approaches such as STRUCTURE (**Pritchard et al., 2000**) and the asymmetric island model (**Wilson et al., 2008**); Bayesian methods, such as sourceR (**Miller et al., 2017**); and hybrid methodologies which can variably include additional information, such as exposure risk (**Ravel et al., 2017**). These methodologies require significant bioinformatic expertise and research time, are computationally expensive, and scale poorly with the increasingly vast collections of bacterial genomes available for analysis.

In order to promote rapid outbreak responses, novel methods are required to translate the increasing volumes of pathogen genomes generated by surveillance programs into immediate and actionable information for epidemiologists. *S*. Enteritidis represents a desirable initial target for such tools due to its large public health burden. WGS provides high-resolution information about pathogen strain relatedness and, by association, contains contextualized information on geographical or host origin. In the case of *S*. Enteritidis, which has a population structure observably stratified by geographical source (**Feasey et al., 2016**; **Li et al., 2021**), there is potential for genomics to provide key information for successful epidemiological follow-up, namely the likely country of origin of an infectious strain, and an opportunity to rapidly enact intervention strategies. Machine learning (ML) approaches have recently shown utility in predicting the original host of infectious *S. enterica* and *Escherichia coli* (**Lupolova et al., 2017**; **Lupolova et al., 2019**; **Wheeler et al., 2018**; **Zhang et al., 2019**), *S. typhimurium* (**Munck et al., 2020**), *Campylobacter jejuni* (**Arning et al., 2021**), and *Listeria monocytogenes* (**Tanui**

*et al., 2022*), but has yet to be applied to the problem of geographical source attribution. Herein we present the first implementation of a hierarchical machine learning (hML) classifier for geographical source attribution of *S.* Enteritidis genomes. We applied this model to the UKHSA's large and uniquely detailed genomic database of clinical *S.* Enteritidis isolates collected between January 2014-April 2019. Using these data, we have generated a fully automated analysis pipeline with which to monitor imported cases of *S.* Enteritidis. The hML model was designed to provide a granular and multi-level prediction of the geographical origin of an *S.* Enteritidis genome in under 4 minutes directly from raw sequencing reads.

## Results

### The UKHSA genomic surveillance program consistently samples *S.* Enteritidis associated with international travel through time

Broad and unbiased surveillance by the UKHSA of all clinically reported *S.* Enteritidis cases in the UK coupled with returning traveler data has provided a large genomic dataset representative of *S.* Enteritidis infections in the UK between 2014–2019 (*Figure 1*). This consisted of 10,223 isolates, of which 3,434 had matched recently reported travel data collected as a part of the UKHSA's 'enhanced surveillance' program.

Recent travel was reported to 122 countries across five continents. A source of potential bias, common to bacterial genomics analysis, is the over-representation of clonally related isolates due to their increased prevalence during outbreaks (*Ebel et al., 2016*; *Feil and Spratt, 2003*). A clone was defined as a single-linkage cluster of isolates with five or less single nucleotide polymorphisms (SNPs), a definition routinely-used by the UKHSA in genomic disease surveillance as part of the 'SNP Address' framework for describing clones (*Chattaway et al., 2019*; *Dallman et al., 2018*). In order to reduce the influence of highly related clonal outbreaks on the resulting ML model, a single, random representative isolate was selected for each clone per country. After quality-filtering, downsampling, and the removal of countries with a low-incidence of reported infections of UK tourists (<10 total isolates), 2,313 genomes from 38 countries from four continents were included in the final dataset for ML (*Figure 1A*).

Grouping these countries by geographic region and subregion using the UN M49 Standard for Regional Codes provided a framework for a granular geographical analysis and an established hierarchy of countries/subregions/regions (*Statistics Division of the United Nations Secretariat, 2020*). Phylogenetic analysis indicated that the dataset displayed a strong phylogeographical signal, with large clusters of isolates from geographically related countries clustering together (*Figure 1B*). An interactive maximum likelihood phylogenetic tree is available in Microreact (*Argimón et al., 2016*). *S.* Enteritidis infection rates in Europe and Asia were highly seasonal, with significantly increased rates of infection during the summer months (*Figure 1C*). This was less pronounced in travel to/from the Americas and Africa.

An analysis of the consistency of sampling effort over time identified that some notable countries in the dataset were composed of samples collected predominantly during a single year, such as Sri Lanka, Tunisia, and Dominica, and others were missing data from one or more years (*Figure 1—figure supplement 1*). However, a comparison of the UKHSA collection to a dataset of location/date-matched genomes from the NCBI Pathogen Detection Database identified that the UKHSA dataset represented a significantly more consistent sampling effort which was less influenced by sporadic outbreaks (*Figure 1D*).

The countries with the highest number of travel-associated *S.* Enteritidis cases were Turkey (804), Spain (357), Egypt (343), Cuba (190), and the Dominican Republic (117) (*Figure 1A*). Controlling for the volume of UK travelers as recorded by the Office of National Statistics over a matched time period allowed for the identification of countries with disproportionate low- or high-risk of *S.* Enteritidis infection after traveling (*Figure 1E*; *Office of National Statistics, 2020*). Travelers to Turkey and Egypt were at a disproportionately higher risk of *S.* Enteritidis infection during this period. Conversely, travelers to France and Spain, two of the more popular travel destinations for UK travelers, were very low-risk. This highlights that the dataset was a product of the volume of UK-travel combined with a variable risk of infection per country. Consequently, there was a large degree of class imbalance (i.e. different number of isolates per country) in the dataset used for ML, in addition to low/absent sampling

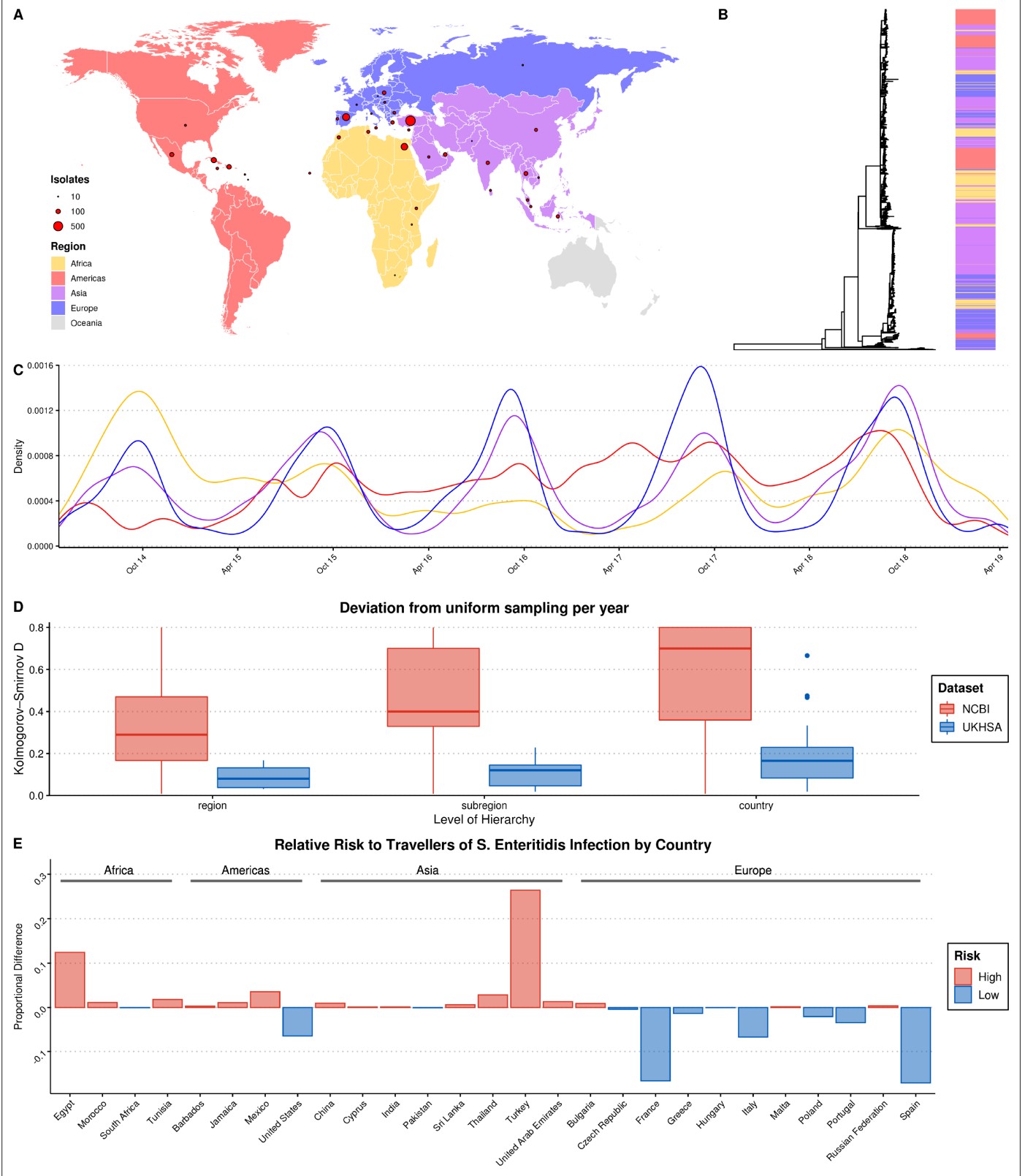

**Figure 1.** Summary of *S*. Enteritidis isolates collected by the UKHSA from UK clinical patients who recently reported foreign travel between 2014–2019. (**A**) Geographical distribution of 2,313 *S*. Enteritidis isolates by reported foreign travel. Variably sized points represent the number of samples per country. The map is colored by region (Africa: yellow, Americas: red, Asia: purple, Europe: blue). (**B**) Maximum likelihood phylogenetic tree of 2,313 *S*. Enteritidis isolates with bar colored by region of origin. (**C**) Kernel density plot indicating sampling density per region through time. No correction was

*Figure 1 continued on next page*

*Figure 1 continued*

made for the seasonal variation observed in international travel. (**D**) Comparison of the consistency of the sampling effort of the UKHSA to all publicly available *S. Enteritidis* isolates on NCBI for the same period. Isolates were resampled to control for variable sample number per year and compared to a uniform distribution using the Kolmogorov-Smirnov D statistic (NCBI = red, UKHSA = blue). Higher values indicate greater deviation from a uniform distribution. (**E**) Relative risk per country of acquiring *S. Enteritidis* infection when traveling. A risk score was generated by dividing the proportion of UKHSA clinical isolates per country by the proportion of all UK travelers traveling to that country as recorded by the Office of National Statistics (ONS). Only countries present in both datasets were used to calculate proportions.

The online version of this article includes the following figure supplement(s) for figure 1:

**Figure supplement 1.** Summary of 2,313 *S. Enteritidis* isolates collected by the UKHSA from UK clinical patients during 2014-2019 who recently reported foreign travel to 38 individual country classes. Each panel contains a bar chart of isolate counts per year per country class.

---

coverage of some parts of the globe. However, when considering larger geographical groupings (i.e. subregion/region) these imbalances were less pronounced or absent.

## A novel hierarchical model provides real-time geographical source attribution prediction directly from sequencing reads in under 4 min

Taking advantage of the hierarchical structure of geographical data, we designed a multi-level hML classifier following a 'Local Classifier per Node' framework (*Silla and Freitas, 2011*). This was made up of 16 individual multi-class classifiers, one per node (one root, four regional, 11 sub-regional). Multi-class classifiers allow for the classification of samples into three or more classes (i.e. Region = Europe, Africa, Americas, Asia), rather than making a series of binary classifiers per individual class (e.g. Africa/Not-Africa, Europe/Not-Europe, etc.). In total, 53 individual classes (four regions, 11 sub-regions, and 38 countries) were included in the model. Sample classification was performed using a top-down approach, where samples are first classified at the root node into 'regions' (e.g. Africa) then, if the predicted probability is greater than a minimum threshold value (0.5), samples are passed to the appropriate regional node to be classified into a nested subset of 'subregions' (e.g. Northern Africa) and finally passed to a subregional node where they are classified into a nested subset of 'countries' (e.g. Egypt) (*Figure 2A*). Using this scheme no samples could be attributed to a class that was not a predicted class of a previous classifier (i.e. Africa→Southern Asia→France is not possible, but Africa→Northern Africa→Egypt is). Sample classification was exclusive, disallowing multiple classifications on the same hierarchical level for a single sample.

For rapid and minimal sample processing and to provide end-to-end sample classification directly from the sequencer, the model was trained on filtered unitig patterns (presence-absence) generated from quality-controlled genomic short-read data files. Each local classifier per node was trained using only data pertinent to that node (e.g. a local subregion classifier was trained only on the data from its constituent countries). This end-to-end process, from FASTQ to sample prediction, is available on GitHub (https://github.com/SionBayliss/HierarchicalML, copy archived at *Bayliss and Cowley, 2023*).

Due to the imbalanced nature of the real-world surveillance dataset, it was necessary to resample the data to generate less biased datasets for training. To this end, a range of classifier and resampler algorithms were compared before selecting the top-performing models (*Figure 2B–C*). Models were assessed using a range of accuracy metrics, although the F1-score (or its hierarchical alternative, hF1) was preferentially used. The F1-score or F-score is the harmonic mean between the precision (the number of true positives divided by the number of true positives plus the number of false positives) and recalls (the number of true positives divided by the number of true positives plus the number of false negatives) and allows for a partial representation of both values in a single metric. The top four models subsequently underwent feature selection (*Figure 2D–E*) followed by parameter optimization using the TPOT genetic algorithm (*Figure 2F*). The optimized hML model produced a more accurate classification of the test dataset than a 'flat' classifier applied to a similarly pre-processed dataset (macro F1: 0.61) (*Supplementary file 1A*). Based on these comparisons, the most desirable assessment metrics overall (i.e. high macro F1 at the country level, *Figure 2E*) were from feature selection by a Random Forest (RF) classifier (25,000 unitig patterns) before random oversampling to correct for class imbalance and final classification using an optimized RF model. Assuming <100 x read coverage, the entire pipeline takes ~3.5 min to classify novel samples using the pre-optimized model.

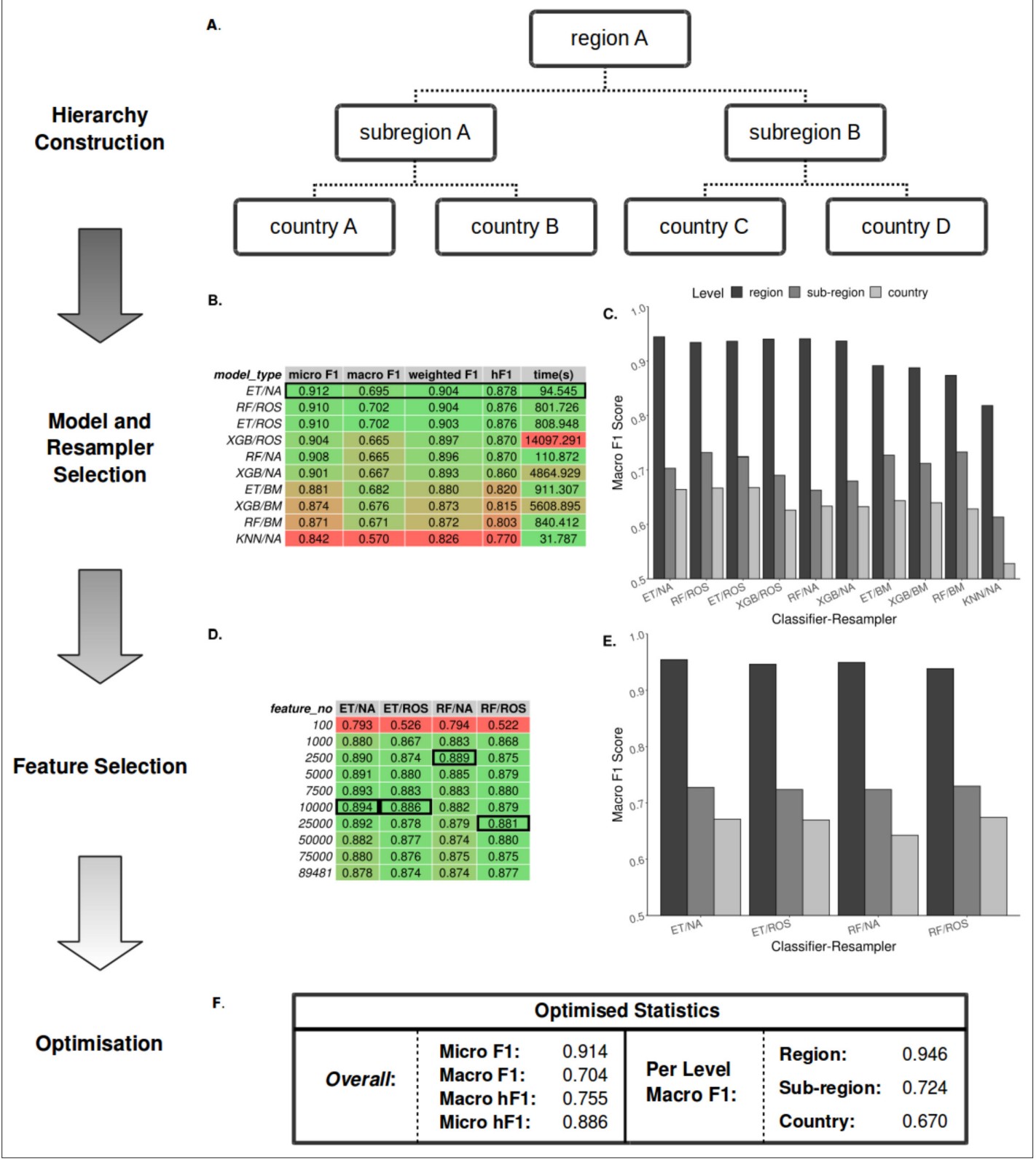

**Figure 2.** Summary statistics showing model and resampling scheme selection, feature selection, and optimization of the *S*. Enteritidis hML source attribution model. (**A**) Example schematic of a geographical hierarchy based upon the UN M49 Standard for regional codes. (**B**) Table of summary statistics for the ten top-performing co-optimized model and resampling methods from a cohort of 36 combinations, sorted by hF1 (high-low). Training time is reported in the final column in seconds. A black box indicates the top four models used for feature selection. (**C**) Grouped bar chart comparing

*Figure 2 continued on next page*

*Figure 2 continued*

macro F1 per hierarchical level for the ten top-performing model/resampled combinations. (**D**) Table of summary statistics for random forest feature selection applied to the four top-performing co-optimized model and resampling methods. Black boxes indicate the optimal number of features per combination. (**E**) Grouped bar chart comparing macro F1 per hierarchical level for the four top-performing co-optimied model and resampling methods after feature selection optimization. (**F**) Summary statistics for the final optimized Random Forest - Random Oversampler model (25,000 features selected). ML Model abbreviations: Random Forest (RF), XGBoost (XGB), Extra Trees (ET), K-Nearest Neighbours (KNN). Resampler abbreviations: No Resampling (NA), Random Undersampler (RUS), Random Oversampler (ROS), Balancing Mean (BM), Hierarchical Mean (HM).

## Granular predictions are provided at a regional, subregional, and individual country level

Classification metrics were highest at the regional level (macro F1: 0.954), but less discriminatory at the sub-regional (macro F1: 0.718) and country levels (macro F1: 0.661) (*Figure 3A*). There was a moderate degree of variation in predictive accuracy between different geographical locations. Africa was the most consistent accurately classified region, with all country and subregional classes presenting a hF1 of >0.7. In The Americas, Latin American and Caribbean countries showed very high classification metrics (hF1: >0.8), whereas samples for the United States were consistently misclassified. All Asian classes were classified at moderate to high accuracy (hF1: 0.58–0.95). Europe was generally well classified (hF1: >0.66), but contained two classes, France and Italy, which were classified poorly (hF1: ~0.3). Further scrutiny of poorly performing classes showed a correlation between a lack of training data and lower prediction accuracy (*Figure 3—figure supplement 1*). This did not fully explain the observed results as some countries with a similarly low number of samples to poorly predicted countries (e.g. Czech Republic, Pakistan) showed moderate classification accuracy. An analysis of genetic diversity within classes indicated that at least two of the most poorly classified countries, France and the United States, displayed both low numbers of samples and high genetic diversity (*Figure 3B*, *Figure 3—figure supplement 2*). Samples from recent travel to France were particularly diverse, arising from multiple highly divergent clades of *S. enteritidis*. A range of countries that were commonly visited by UK travelers, such as Cuba, Egypt, Indonesia, Jamaica, Malta, Spain, Thailand, Tunisia, and Turkey, were well predicted (hF1: >0.9).

This model allows for a separate measure of predictive confidence (i.e. the probability value reported by the appropriate classifier) to be reported to the end-user for each hierarchical level. An example of how this uncertainty maybe visualized for the end-user is available in *Figure 3C* and mirrored in the interactive Microreact project. These figures include three columns containing the predicted class of each sample per hierarchical level and three matched columns with the probability of the prediction per class as reported by the appropriate classifier. This also allows for a user to visually inspect the variable level of predictive confidence for each bacterial lineage within its particular phylogenetic context. For ease of interpretation, an alternative to *Figure 3* has been included in the supplementary information (*Figure 3—figure supplement 3*) which displays the % of isolates of each class correctly assigned to that class instead of HF1.

In summary, these results suggest that the optimized model can attribute the geographical source of *S. Enteritidis* isolates with high confidence at a regional level whilst also providing more nuanced and granular predictions for a range of countries regularly visited by UK travelers with a very high degree of accuracy. The resulting measures of confidence in individual sample predictions can be easily communicated to end-user an a simple and easily interpretable manner.

## Models demonstrate durability to future predictions with 2 years previous training data providing the sufficient signal for accurate subsequent year predictions

Bacterial population lineage composition is not expected to remain static through time, therefore, predictive models based on genomic data require periodic retraining. To understand the amount of data required for accurate prospective prediction, we compared the outcomes of four yearly window sizes (1, 2, 3, and 4 years) for the prediction of subsequent years (*Figure 4A*). Predictive accuracy and consistency of prediction of the subsequent year improved on increasing window size, with hF1 beginning to asymptote after 2 years. The largest improvement was observed between 1–2 years' worth of data. A minor decrease in predictive accuracy (micro F1: ~0.05 per year) was observed for

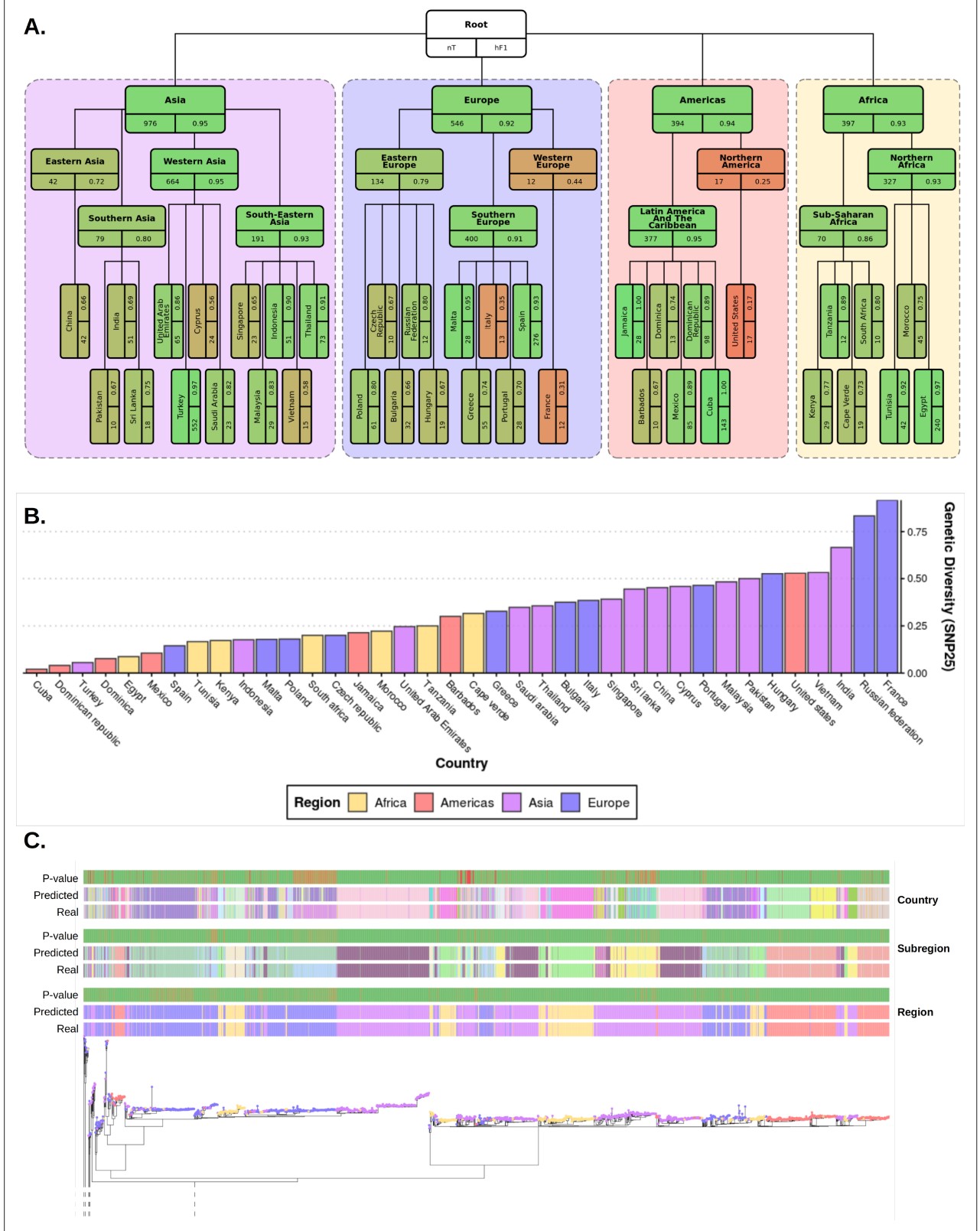

**Figure 3.** Plots summarizing results from hierarchical machine learning (hML) model applied to the *S*. Enteritidis test dataset and a measurement of the genetic diversity of *S*. Enteritidis sample collection per country. (**A**) Diagrammatic representation of classification metrics of the hML classifier applied to the test dataset. Links between classes/nodes in the hierarchy are indicated by connecting lines. Boxes represent individual classes in the Machine learning (ML) model and are colored by their hierarchical F1 (hF1) scores. The top panel of each class box displays the class label, the bottom left panel

*Figure 3 continued*

indicates the total number of samples for that class before the train/test split (75%/25%) and the bottom right panel shows the class hF1 score calculated from the test dataset. Classes within individual 'regions' (continents) were contained in a colored background panel. (**B**) Bar plot of genomic diversity per country. Genomic diversity was estimated as the number of 25 single nucleotide polymorphism (SNP) single linkage clusters divided by the total number of samples per class. Panels (**A**) and bars (**B**) were colored according to region (Africa: yellow, Americas: red, Asia: purple, Europe: blue). (**C**) Maximum-likelihood phylogenetic tree of the UKHSA *S.* Enteritidis isolates from the training and test datasets. The real label and predicted label (each assigned a unique color) of each isolate alongside the p-value generated by the model is shown for each hierarchical class level - region, subregion and country. Colored tree tips represent the region class of each isolate. An interactive version of the tree and associated metadata are available in Microreact.

The online version of this article includes the following figure supplement(s) for figure 3:

**Figure supplement 1.** Scatter plots summarizing the number of samples (x-axis) vs hF1 score (y-axis) generated by the optimized hierarchical machine learning (hML) model.

**Figure supplement 2.** Scatter plots summarizing hF1 score (x-axis) vs. the number of samples/genomic diversity at SNP5/genomic diversity at SNP50 (y-axis – top/middle/bottom).

**Figure supplement 3.** Diagrammatic representation of classification metrics of the hierarchical machine learning (hML) classifier applied to the *S.* Enteritidis test dataset showing % isolates correctly assigned per class.

each additional year into the future (***Figure 4B***). A regional breakdown of the hF1 score indicated that a one-year window varied in predictive accuracy per region per year (***Figure 4C***), but that a two-year window provided more consistent and accurate predictions (***Figure 4D***). These results indicate that should this model be instituted for ML-enhanced genomic surveillance, it would require retraining on the two previous years samples each year to provide an optimal trade-off between predictive accuracy and training time.

## The optimized hML model provides accurate predictions in 4/5 validation datasets

The hML model was further validated by application to a series of external, independent datasets (***Supplementary file 1D***). The initial datasets were from two UK-based, well-characterized, and epidemiologically-traced imported food outbreaks. Sample redundancy between validation

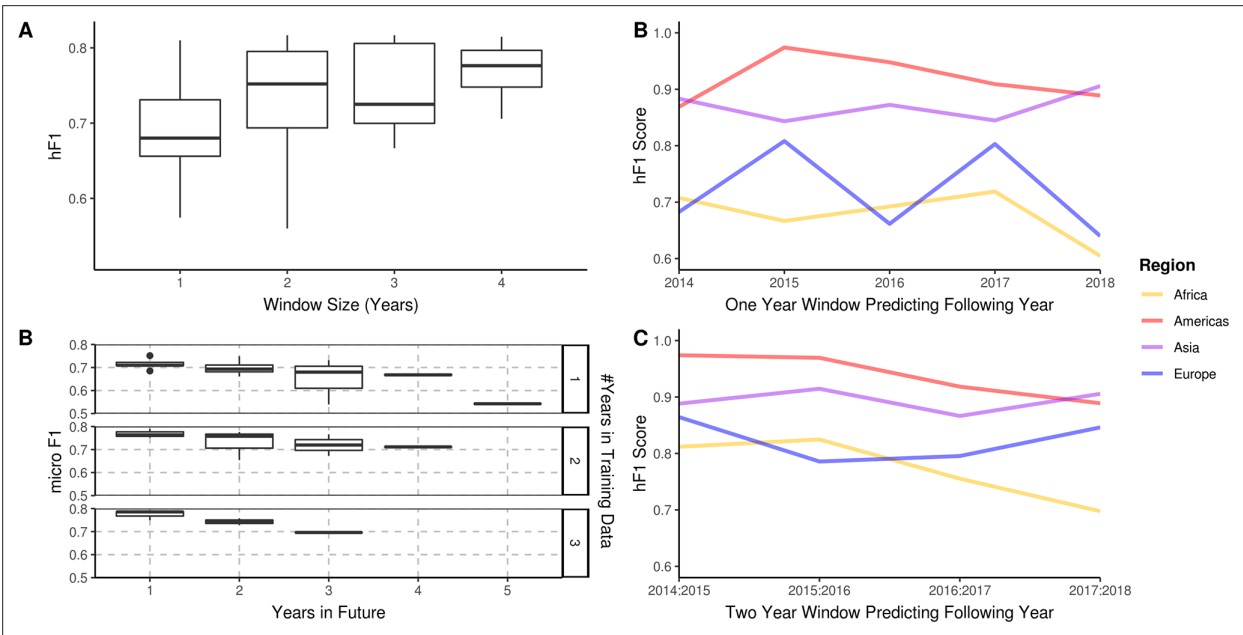

**Figure 4.** A longitudinal analysis of the predictive performance of hierarchical machine learning (hML) models on 2,313 *S.* Enteritidis samples. (**A**) The hF1 scores of hML Random Forest (RF) models trained on a subset of samples from a 1–4 year moving window predicting the following year. (**B**) The micro F1 scores of hML RF models trained on a subset of samples from a 1–3 year moving window predicting 1–5 years into the future. (**C**) Regional hF1 scores of hML RF models trained on one-year sampling windows predicting the following year. (**D**) Regional hF1 scores of hierarchical models trained on two-year sample windows predicting the following year.

and training datasets was removed before comparison. First, a 2015 outbreak epidemiologically-traced to eggs imported from Spain (*Inns et al., 2017*) was 100% correctly attributed to a Spanish origin (*Figure 5A*). Second, a multi-country outbreak originating from Polish egg farms (*Pijnacker et al., 2019*), comprising of two distinct lineages each differing by five or fewer SNPs, was correctly attributed to a European origin (131/131 cases), but subsequently misattributed to a Southern Europe (131/131 cases), Spanish origin (103/130) (*Figure 5B*). This complex outbreak was a particularly difficult test case for the model, as it had been continuously causing cases in 16 European countries for several years (2015–2018). The outbreak cases were also phylogenetically distinct from those associated with travel to Poland in the UKHSA dataset (*Figure 5B*). It was possible to identify samples attributed to the outbreak in our training data, this included six samples that were present in the training dataset. These six samples were labeled as having been acquired from Portugal (3), Spain (1), Greece (1), and Turkey (1) based on recently reported travel information, despite having got their infection from Polish eggs. 5/6 of these were of a Southern European origin, and likely contributed to the misattribution of the samples from the Polish validation datasets to a Southern European origin.

The model was further tested on datasets generated by other non-UK institutions. Three countries from three different regions were identified from the NCBI database as having acceptable sample numbers (>20), falling within the time-frame of the current model (2014–2019), and having been sampled from a country included in the current model hierarchy (South Africa, Singapore, and Poland). The Polish dataset, sampled from poultry, was attributed to a European origin with high accuracy (34/35, 97.1%), 19 of these were subsequently attributed to an East European origin (19/35, 54.3%) or which 18 were attributed to Poland (18/35, 51.4%) (*Figure 5C*). The remaining 15 samples were misclassified as having a Spanish origin (15/35, 42.9%). The South African dataset of clinical cases (*Smith et al., 2020*), was correctly attributed to a South African origin with 100% accuracy (25/25) (*Figure 5D*). The Singaporean dataset of human outbreaks (*Octavia et al., 2018*) was correctly attributed to South-East Asia (48/48) with 91.7% of samples being correctly attributed to a Singaporean origin (44/48) and 4 samples being attributed to Indonesia (2), Malaysia (1), and Thailand (1) (*Figure 5E*). An additional visualization of the validation datasets alongside the training and test datasets has been presented in *Figure 5—figure supplement 1*.

Application of the hML model to independent validation datasets indicated that the model provided highly accurate and granular predictions for single-country outbreaks which occur over short time-frames. A large-scale, long-term, multi-country outbreak was poorly predicted by the model, likely due to a small number of outbreak isolates present in the training dataset being incorrectly labeled as having a Southern European origin. However, the application of the hML model to a range of other independent validation datasets provided highly accurate and granular predictions.

## Discussion

Outbreaks caused by foodborne pathogens, where rapid response times are essential for effective interventions, represent an epidemiological challenge for public health bodies as they arise from complex, interconnected global supply chains which include many potential sources of infection. Our optimized hML model generates accurate predictions of the geographical origin of *S. Enteritidis* genomes directly from raw read data in under 4 min per sample. The output of this pipeline is a predicted probability per hierarchical level (i.e. region, sub-region, and country levels) allowing for granular source attribution alongside an assessment of confidence in the prediction. The observed classification accuracy of the model was high, varying across both hierarchical levels and individual classes (*Figures 2–3*). Accuracy was highest at the regional level (macro F1: 0.954), which contained the largest number of samples for training and the lowest level of class imbalance, before dropping at the subregional (macro F1: 0.718) and country levels (macro F1: 0.661), both of which contained generally smaller and more variable numbers of samples, increasing class imbalance (*Figure 3*). It should be noted that these macro values, being an average of the F1 score across all classes, were strongly influenced by outliers representing a handful of classes with poor classification accuracy (e.g. United States, France, US, Italy). Classification accuracy was negatively associated with both low sample numbers and increasing within-class genetic diversity (*Figure 3—figure supplement 1*, *Figure 3—figure supplement 2*), although this did not fully explain the variation in predictive accuracy observed in the model, suggesting other complex factors may be involved. Although variation in predictive accuracy was observed, a number of commonly visited countries were predicted with extremely high

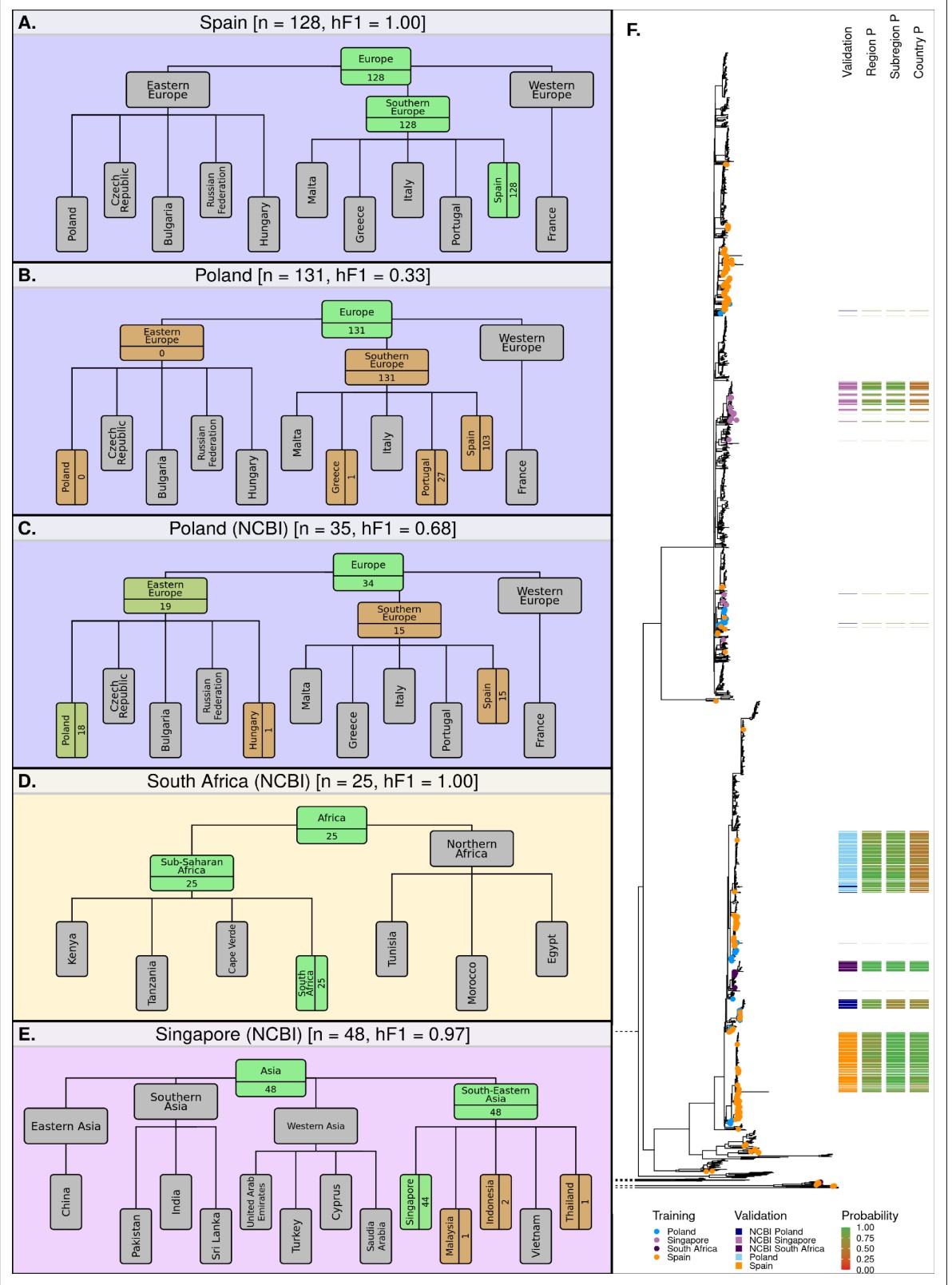

**Figure 5.** Classification summaries of the optimised hML model applied to five additional independent validation collections of *S.* Enteritidis genomes. (**A**) 128 samples from an international outbreak that originated in Spain in 2015 (*Inns et al., 2017*). (**B**) 131 samples from a large-scale international outbreak that originated from Polish eggs between 2015–2018 (*Pijnacker et al., 2019*). (**C**) 35 samples from Poland were uploaded to the NCBI database between 2014–2019. (**D**) 25 samples from South Africa were uploaded to the NCBI database between 2014–2019 (*Smith et al., 2020*). (**E**) 48

*Figure 5 continued on next page*

*Figure 5 continued*

samples from Singapore were uploaded to the NCBI database between 2014–2019 (*Octavia et al., 2018*). The number of samples assigned per class is indicated for classes relevant to the query dataset. Class boxes are colored by the proportion of correctly/incorrectly classified samples (correct: green, incorrect: red). (F) Maximum-likelihood phylogenetic tree of *S*. Enteritidis isolates from the training, test, and validation datasets. Validation dataset isolates are displayed as a colored bar alongside the p-value of the model predictions for associated region, subregion, and country class. Colored tree tips represent the training isolates from the four countries tested across the validation datasets.

The online version of this article includes the following figure supplement(s) for figure 5:

**Figure supplement 1.** Maximum-likelihood phylogenetic tree of *S*. Enteritidis isolates from the training, test, and validation datasets.

predictive accuracy (e.g. Cuba, Egypt, Indonesia, Jamaica, Malta, Spain, Thailand, Tunisia, and Turkey) which would be of high-utility for the UK epidemiologists tracing foodborne outbreaks.

Selection and optimization of the ML classifier and resampler methods, as well as pre-classification feature selection, were performed in a stepwise manner to assess the impact of each step on the resulting model. Ensemble classifiers outperformed other classifiers (*Figure 2*, *Supplementary file 1B*). Interestingly, considering the large variation observed in some nodes of the hierarchy, the class imbalance was not found to have a large impact on the resulting model. Two of the top four best-performing classifier/resampler combinations included no resampling. These were both ensemble classifiers, suggesting that these were better able to account for the high dimensionality, high collinearity, and variable levels of class imbalance than other classifier models. The resampling itself was observed to differentially impact the various hierarchical levels (*Figure 2*). The model required a sample to be classified at a higher hierarchical level before being passed on to the next nested level (e.g. classified as European before being classes as West/East/South European) which biased model selection towards favoring classifiers which improved regional and subregional predictions. It should be noted that each classifier/resampler/feature selection step was applied uniformly across the hierarchy prior to parameter optimization, to allow for an assessment of individual steps on the resulting model. An attractive, although less informative, approach for future retraining would be to optimize all steps (classifier/resampler/feature selection) on a per-node basis using an AutoML method, such as the genetic algorithm used in the current work to optimize hyperparameters in the final hML RF model (*Olson et al., 2016*). Validation using external datasets indicated that the model robustly predicted the majority of novel samples, but was less effective for complex multi-country, multi-year outbreaks (*Figure 5*).

We acknowledge that this study has a number of important limitations. Whilst we have presented evidence that the hML model can robustly predict geographical sources for *S*. Enteritidis, the COVID-19 pandemic has resulted in a vast disruption to international travel over the past 2 years. The UKHSA is already seeing a return to pre-pandemic levels of imported infection and expect the pattern of the seasonality of imported *S*. Enteritidis infections to return. However, there may be unquantified impacts on the global food network which could result in a reduction in the prediction accuracy of the current model and would likely require retraining on post-pandemic data. Furthermore, the dataset, collected as part of the UKHSA's national genomic surveillance program, is a product of two factors; the countries to which UK residents commonly travel and the variable likelihood of *S*. Enteritidis infection whilst in these countries (*Figure 1*). Both factors influence the geographical distribution of sampling locations present in the dataset as well as the number of available samples per country. For example, of the 122 countries present in the initial dataset only 38 (31%) were present in sufficient quantities over the surveillance period to include in the model. The second factor influencing the composition of the dataset is the variable risk of infection posed by the countries present in the dataset. Some commonly visited countries posed only a minor risk of infection (e.g. France) whilst others posed a disproportionately high risk of infection on visiting (e.g. Turkey) (*Figure 1E*). The risk of infection per country is likely to be a product of the overall prevalence of *S*. Enteritidis as well as the local circulation of high-virulence or low-virulence strains, infections with which is more or less likely to result in referral to the clinical setting. Due to the complex nature of both the global food supply chain and pathogen transmission dynamics, it is likely that these factors will cause misattribution in some cases; for instance, UK-travelers might travel to country A and consume contaminated foodstuffs imported from country B which was absent from the training data, and the model would then 'correctly' assign future cases to country A. One might also imagine that country C, which is only rarely visited by UK-travelers, produces a contaminated foodstuff and exports it to country D,

which is often visited by UK-travelers, causing the model to misattribute a sample from country C to D. In these cases, the model would provide useful information on the origin of infection but would not ultimately identify the true reservoir of the pathogen. Evidence of this form of misattribution has been presented in the Polish eggs outbreak dataset used for validation (*Figure 5B*). This outbreak represents a particularly difficult attribution problem as many of the contaminated eggs were distributed to 18 different EU countries during the sampling period used to train the model – allowing for the inclusion of a small number of erroneously labeled samples in the training data and resulting in inaccurate classification. This outbreak represents an important limitation of the current model and illustrates how multi-country outbreaks or long-established food network contamination pathways can confound accurate source attribution.

Other potential forms of mislabeling in reported travel data include the incorrect reporting of multi-country trips or infection of travellers during return-trip layovers. The ONS records travel destination as the primary location visited, so no estimates of the proportion of samples attributable to multi-country trips could be made, although they will certainly represent a minority of the data. Recent surveys have estimated that layovers occur on only a small proportion of flights (~15%), although this could potentially have a larger impact on the data due to the bias in detecting cases that occur towards the end of a trip, which could result in a patient being admitted to UK-based clinical care setting, rather than one overseas (*Department for Transport, 2018*; *Department of Transport, 2018*). Another limitation of this work was its reliance on recently reported travel data, which was used as a proxy for the prevalence of infections arising from imported foodstuffs. Whilst there is evidence that this approach can capture highly-relevant disease strains present in foodstuffs imported to the UK (*Chan et al., 2023*; *Chattaway et al., 2023*; *Jenkins et al., 2019*), strain prevalence is likely to vary across national borders due to differences in national and domestic regulations and food-processed methods.

We have presented a conceptually simple hML model which is able to successfully predict the geographical source of infection for a wide range of popular destinations frequented by UK travelers. However, the basis of this work was a single dataset which, although exceptional in the breadth of sampling and metadata, compositionally reflects locations routinely visited by UK travelers. A number of destinations were present in the dataset for which there is little to no data present in public databases during a matching time period (e.g. Malta). This demonstrates the additional utility provided by the collection of recent travel information as part of a national surveillance program to generate consistent geographical coverage for disease-sampling over a much larger geographical area. Travel-associated samples have recently been used to improve our understanding of the geographical distribution of various *S.* Typhi lineages by being used as an informal substitute for sampling efforts from geographical regions lacking formal disease surveillance programs (*Carey et al., 2022*). One could imagine that coordination between public health bodies with complementary citizen travel preferences would allow for a small network of national genomic surveillance initiatives to provide global coverage of gastrointestinal diseases without the costly necessity of instituting comprehensive global surveillance initiatives. If this was coupled with a similar hML model to the one detailed here, then rapid and precise global predictions of the source of gastrointestinal disease outbreaks could be achieved.

The model presented herein is conceptually simple and does not incorporate temporal information, account for seasonal fluctuations in disease incidence, allow for multiple labels to be assigned to samples, or incorporate animal host/food source information, which may improve the prediction of multi-country outbreaks or provide additional information to further enhance epidemiological follow-up (*Lupolova et al., 2017*; *Lupolova et al., 2019*; *Wheeler et al., 2018*; *Zhang et al., 2019*). Furthermore, the UK was excluded as a potential geographical source in the model due to the extremely low prevalence of *S.* Enteritidis in domestic poultry flocks and a lack of clinical isolates linked to consumption of locally produced foodstuffs (*European Food Safety Authority and European Centre for Disease Prevention and Control, 2017*). Future studies could include samples collected via veterinary or food chain-based surveillance programs to provide finer-scale inference of causation or provide model coverage of low-prevalence geographical locations.

This study provides a framework for future hML applications in the area of pathogen genomics. Other geographically stratified problems which might benefit from a hierarchical approach include antimicrobial resistance in *E. coli* (*Ingle et al., 2018*), the transmission of *Staphylococcus aureus* in

hospital networks (*Donker et al., 2017*), and application to other serovars of *S. enterica* (*Lupolova et al., 2017*; *Munck et al., 2020*). This work represents the first application of hML for the automation of genomics-based geographical source attribution. The rapidity of predictions from raw sequencing data should greatly enhance epidemiologists' ability to trace the source of gastrointestinal outbreaks by condensing complex genetic data into understandable and actionable outputs.

## Methods

### Genome collection and processing

The initial dataset consisted of 10,223 *S.* Enteritidis isolates collected and sequenced by UKHSA between 2014–2019 as a part of their routine disease monitoring program. Raw read data were downloaded from the Sequence Read Archive (Bioproject: PRJNA248792) (*Supplementary file 1C*). Reads were filtered using Trimmomatic 0.39, residual Illumina adapter sequences were removed, the first and last three base pairs in a read were trimmed, before a sliding window quality trimming of four bases with a quality threshold of 20 were applied, and reads of less than 36 bp (after trimming) were removed (ILLUMINACLIP:PE_All.fasta:2:30:10:2:keepBothReads SLIDINGWINDOW:4:20 LEADING:3 TRAILING:3 MINLEN:36) (*Bolger et al., 2014*). The coverage of the resulting file was estimated against the size of *S.* Enteritidis reference genome P12510 (*Thomson et al., 2008*) and downsampled to ~100 x coverage using in-house scripts. Musket 1.1 was applied for k-mer spectrum read correction using a k-mer size of 31 bp. Unitigs were generated directly from filtered reads using bcalm 2.2.3 with a k-mer size of 31 bp and a minimum k-mer abundance of six (estimated from data) (*Chikhi et al., 2016*).

Reads were mapped to *S.* Enteritidis reference genome P12510 and variants were called using SNIPPY 4.6 with a minimum coverage of 10x and a mapping quality of 60 (*Seemann, 2021*). A reduced alignment of variant sites was generated using snp-sites (*Page et al., 2016*) and passed to IQ-Tree for phylogenetic reconstruction using a GTR + I + G substitution model and ultra-fast bootstrapping (1000 bootstraps) (*Nguyen et al., 2015*). An interactive maximum likelihood phylogenetic tree is available in Microreact.

### Isolate selection for Machine learning

Of the initial 10,223 isolates, 3,434 had matched recent reported travel data, in the form of 'country' of recent travel (within 28 days of generating symptoms), provided by the clinical laboratories on submission of the isolate. These isolates were selected as potential candidates for ML model construction and testing. The 3,434 initial isolates were subsequently filtered to identify samples that had both consistent metadata and good sequence quality. The criteria for inclusion included: (a) clear and uncontradictory recent travel metadata, unrecognized or invalid regions (e.g. Yugoslavia) were removed; (b) reads that were not genetically distant from the majority of isolates in the collection as measured by MASH distance; (c) reads that had >28 x coverage of *S.* Enteritidis reference genome P12510; (d) samples that did not have a total unitig length greater than 5,250,000 bp; (e) k-mers created from the reads that did not have a singleton frequency of >0.5. A total of 220 samples were excluded due to these criteria.

Further dataset reduction was performed to remove genetically identical, or near-identical, isolates from the collection prior to processing for ML. SNP Address was used as a proxy for genetic relatedness and a single example of each SNP Address (SNP5) was randomly selected per country for further analysis. Samples were assigned to region and subregion based upon the UN M49 Standard for regional codes (https://unstats.un.org/unsd/methodology/m49/overview) [Accessed: 30 November 2021]. Recent travel information which represented historically distinct subregions or autonomous subregions of a larger country grouping (e.g. Hong Kong), was considered a part of the larger entity for the purpose of classification. Any country with less than ten representative samples after filtering was excluded from further analysis. The final sample collection after filtering contained 2,313 samples (*Supplementary file 1C*).

The relative risk of acquiring *S.* Enteritidis infection when traveling was estimated using the ratio of the proportion of UKHSA clinical isolates reported as having recently traveled to each country over the proportion of all UK travelers traveling to that country as recorded by the (*Office of National*

*Statistics, 2020*). The data used for this comparison was the date (year) and location (country) matched before proportions were calculated.

The NCBI Pathogen Genome database was interrogated to identify *S.* Enteritidis isolates collected in countries present in the hML model over a matching time period [Accessed: 18 November 2021]. 2019 was excluded as UKHSA samples were only available for a relatively short duration of the year (January-April). Variation in the isolate number collected each year was controlled for by resampling with replacement (1,000 samples). The resulting yearly sampling data was compared to a uniform distribution using the Kolmogorov-Smirnov D statistic to quantify deviation from a consistent sampling scheme for both the UKHSA and NBCI reference datasets.

## Unitig processing for ML

A colored De Bruijn graph was constructed for the complete genome collection by passing the unitigs for individual samples to unitig caller v1.2.0 (*Holley and Melsted, 2020*). This identified 426,647 unique unitigs present in the 2,313 genome collection. The resulting De Brujin graph was then queried to establish the presence/absence of each unitig on a per-sample basis. Unitigs present/absent in only a single sample were removed. Unitigs that co-occurred in the same subset of samples (i.e. perfect correlation) were clustered into a single feature 'pattern' for input into feature selection and ML model-building algorithms. This reduced the input from 426,647 unitig features into 94,865 pattern features. The dataset was split into 75–25% train-test ratio stratified by the country for downstream applications.

## Hierarchical classifier design

A class-prediction top-down hierarchical classifier framework was developed to be compatible with any scikit-learn classifier which generates a per-sample predicted probability value. The framework followed a Local Classifier per Node (LCN) approach as detailed in *Silla and Freitas, 2011*, wherein a multi-class classifier was fitted at each node of the hierarchy to differentiate between the various child classes of that node. The geographical hierarchy used as the basis of the classifier was constructed using the three levels of geographical labels, region->subregion->country, assigned by UN M49 Standard for regional codes (*Statistics Division of the United Nations Secretariat, 2020*). At each node, samples were relabeled according to the current hierarchical level (e.g. Cuba and the United States would be labeled as Americas at the root/regional node) before being passed to the classifier. Only samples in classes relevant to the current hierarchical node classifier were included in the training data for each classifier (i.e. samples from countries that were not a part of the region/subregion being trained were excluded).

## Hierarchical classification strategy

The hierarchical classifier framework allowed for the flexible assignment of samples to a single unambiguous class in the hierarchy (i.e. a single region/sub-region/country or unclassified). Samples were first classified at the root (regional) node. A classification was assigned if the predicted probabilities adhered to the threshold criteria, and the maximum probability value at that node exceeds a threshold value (0.5). Only then would the sample be passed to the next node in the hierarchy. This is not permissive of multiple, conflicting classifications.

A range of hierarchical and non-hierarchical statistics was applied to aid model assessment. Standard non-hierarchical statistics were generated for each individual classifier/node (precision, recall, micro/macro/weighted F1) as well as their hierarchical analogs (hP, hR, hF1). These were calculated as described by *Kiritchenko et al., 2005*:

$$hP = \frac{\sum_i \left| \hat{P}i \cap \hat{T}i \right|}{\sum_i \left| \hat{P}i \right|} \qquad hR = \frac{\sum_i \left| \hat{P}i \cap \hat{T}i \right|}{\sum_i \left| \hat{T}i \right|} \qquad hF = \frac{2 * hP * hR}{hP + hR}$$

where $P^i$ is the set of predicted classes consisting of the most specific class (i.e. the lowest level of the hierarchy) predicted for test example $i$ plus all parent classes and $T^i$ is the set of classes consisting

of the most specific true class of test example *i* and all its ancestor classes. Summations were calculated over all test samples. Macro hF1 was calculated by taking the average F1 of all classes of interest.

## Feature selection, resampling, model testing, and optimization

During model selection, various combinations of classifier, resampler and feature selection models were applied to the test-train dataset to assess their suitability for model building. These were assessed based on a combination of non-hierarchical statistics including overall micro/macro F1 and micro/macro F1 per hierarchical level. The implemented classifier models included K-Nearest Neighbours, Support Vector, Random Forest, Gaussian Naive Bayes, XGBoost, Extra Trees. All classifier models were implemented using scikit-learn using a set seed value and default parameters with the exception of Random Forest, Extra Trees, and XGBoost classifiers which were run with n_estimators = 1000 and SVC which was run with probability = True.

The implemented resampling schemes included downsampling (smallest class), upsampling (largest class), resampling to the mean count of all classes, and hierarchically aware implementation for the previously described samplers. Hierarchically aware resampling was developed using in-house scripts to iteratively apply a resampler to each of the lowest levels of the hierarchy (country) before passing the resampled data to higher levels in the hierarchy for further resampling.

An all-vs-all comparison of classifier vs resampler models was used to identify the most suitable combinations for further optimization (*Supplementary file 1B*). In all cases, a fixed seed value was used for the comparison of models and resamplers. The top four combinations of classifier-resampler were selected for feature selection comparison. The implemented feature selection method was Random Forest using varying numbers of patterns as training data (*Figure 2D*).

The final classifier-resampler-selection combination was passed to a genetic algorithm framework (TPOT) to identify an approximation of optimal parameters from a wide range of possible combinations (sklearn.ensemble.RandomForestClassifier': 'n_estimators': [100, 500, 1000], 'criterion': ['gini', 'entropy'], 'max_features': np.array([0.05, 0.1, 0.15, 0.2, 0.25, 0.3, 0.35, 0.4, 0.45, 0.5, 0.55, 0.6, 0.65, 0.7, 0.75, 0.8, 0.85, 0.9, 0.95, 1. ]), 'min_samples_split': range(2, 21), 'min_samples_leaf': range(1, 21), 'bootstrap': [True, False]) (*Olson et al., 2016*). The TPOT genetic algorithm used the macro F1 score per node as the optimization metric, was run for 100 generations with a population size of 50, and stratified threefold cross-validation of the input database and was stopped if no model improvement was found for 10 generations. A 'flat' model was also trained and tested in the same manner for comparison, whereby a multiclass Random Forest classifier was provided with a randomly oversampled dataset which only included 'country' class labels (i.e. region/subregion were ignored) (*Supplementary file 1A*).

## Validation dataset collection and processing

Various public datasets were used as additional validation data including outbreaks described in previous publications (*Inns et al., 2017*; *Octavia et al., 2018*; *Pijnacker et al., 2019*; *Smith et al., 2020*) and samples identified from the NCBI Pathogen Genome database [Accessed: 18 November 2021]. In the case of samples taken from previous publications, accession numbers were identified from these manuscripts, and samples were downloaded and passed through the genome and unitig processing pipelines described above. Additionally, NCBI Pathogen Genome database metadata was downloaded [Accessed: 18 November 2021] and filtered to return only samples that had publicly accessible read files, and country metadata from the 38 countries and were collected between 2014–2018. Three representative countries were chosen to trial the model on (Poland, South Africa, and Singapore). Read data was downloaded and passed through the unitig processing pipeline as described above. The presence of unitigs generated from the UKHSA collection which formed the basis of the hML model was ascertained using unitig-counter (*Jaillard et al., 2018*, *Lees et al., 2018*). The unitig features were then converted into the patterns generated from the UKHSA collection as described above.

## Acknowledgements

We would like to acknowledge both Dr. Harry Thorpe and Dr. Nicola Coyle who have both previously contributed to the development of scripts that underlie the unitig processing pipeline. This work was funded by an Academy of Medical Sciences Springboard grant (SBF005\1089). CJ, TD,

and MAC are affiliated to the National Institute for Health Research Health Protection Research Unit (NIHR HPRU) in Gastrointestinal Infections and Genomics and Enabling Data at the University of Liverpool and University of Warwick, respectively in partnership with the UK Health Security Agency (UKHSA). CJ and MAC are based at UKHSA. The views expressed are those of the author(s) and not necessarily those of the NIHR, the Department of Health and Social Care, or the UK Health Security Agency.

## Additional information

### Funding

| Funder | Grant reference number | Author |
|---|---|---|
| Academy of Medical Sciences | SBF005\1089 | Lauren A Cowley |

The funders had no role in study design, data collection and interpretation, or the decision to submit the work for publication.

### Author contributions

Sion C Bayliss, Resources, Data curation, Software, Validation, Investigation, Visualization, Methodology, Writing – original draft, Writing – review and editing; Rebecca K Locke, Data curation; Claire Jenkins, Marie Anne Chattaway, Data curation, Project administration, Writing – review and editing; Timothy J Dallman, Data curation, Supervision; Lauren A Cowley, Conceptualization, Supervision, Writing – original draft, Project administration, Writing – review and editing

### Author ORCIDs

Sion C Bayliss http://orcid.org/0000-0002-5997-2002
Lauren A Cowley http://orcid.org/0000-0003-0835-7971

### Decision letter and Author response

Decision letter https://doi.org/10.7554/eLife.84167.sa1
Author response https://doi.org/10.7554/eLife.84167.sa2

## Additional files

### Supplementary files

• Supplementary file 1. Supplementary Tables. (A) Summary of *S.* Enteritidis isolates collected by the UKHSA from UK clinical patients who recently reported foreign travel to 38 individual country classes. Each panel contains a bar chart of isolate counts per year per country class. (B) Summary of the all-vs-all comparison of classifier vs resampler models used to identify the most suitable combinations for feature selection and additional optimization. The implemented resampling schemes included downsampling (smallest class), upsampling (largest class), resampling to the mean count of all classes, and hierarchically aware implementation for the previously described samplers. Hierarchically aware resampling was developed using in-house scripts to iteratively apply a resampler to each of the lowest levels of the hierarchy (country) before passing the resampled data to higher levels in the hierarchy for further resampling. Abbreviations; RF: Random Forest, KNN: K Nearest Neighbour, SVM: Support Vector Machine, GNB: Gaussian Naïve Bayes, and XGB: Xtreme Gradient Boosting. (C) The 2313 *Salmonella* Enteritidis sample collection was used for training and testing of the hML classifier presented in the main manuscript. All samples were collected by the UKHSA as a part of their genomic surveillance initiative. Recently recorded travel was collected as a part of the UKHSA's 'enhanced surveillance' program. The columns include: SRA accession code, receipt date of isolate, SNP Address, reported country of travel, subregion, and region that the country lies within based on the UN M49 Standard for regional codes. (D) Validation dataset genomic collection and associated metadata.

• MDAR checklist

## Data availability

The final optimised hierarchical model as well as a pipeline for pre-processing raw read data to unitigs/patterns for input and paper data is available from https://github.com/SionBayliss/Hierarchi-calML (copy archived at *Bayliss and Cowley, 2023*) with a short description and tutorial for ease of use. This end-to-end process, from FASTQ to prediction, is open access and available to users under GNU GPLv2 licence . This depository also includes the preprocessed unitig datasets and resulting predictions. Short read sequencing data is available from the Sequence Read Archive (Bioproject: PRJNA248792). Please note that the sequence data has been previously deposited/published in the Sequence Read Archive by PHE/UKHSA and was not generated for this project.

The following previously published datasets were used:

| Author(s) | Year | Dataset title | Dataset URL | Database and Identifier |
|---|---|---|---|---|
| UKHSA | 2018 | Public Health England - Salmonella | https://www.ncbi.nlm.nih.gov/data-hub/genome/?bioproject=PRJNA248792 | NCBI BioProject, PRJNA248792 |
| Smith AM, Tau NP, Ngomane HM, Sekwadi P, Ramalwa N, Moodley K, Govind C, Khan S, Archary M, Thomas J | 2020 | Whole-genome sequencing to investigate two concurrent outbreaks of Salmonella Enteritidis in South Africa, 2018 | https://www.ncbi.nlm.nih.gov/bioproject/PRJNA646325 | NCBI BioProject, PRJNA646325 |
| Octavia S, Ang MLT, La MV, Zulaina Siti, Tien WS, Han HK, Ooi PL, Cui Lin, Lin RTP, Zul AS | 2018 | Retrospective genome-wide comparisons of *Salmonella enterica* serovar Enteritidis from suspected outbreaks in Singapore | https://www.ncbi.nlm.nih.gov/bioproject/PRJNA434621 | NCBI BioProject, PRJNA434621 |

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
