## [Editor Report]

This important study presents a machine learning-based classifier that can accurately determine the geographic origin of a *Salmonella enterica* sample from its whole-genome sequencing data in under five minutes leading to actionable public health insights. Applying the method to 2,313 whole genome sequences collected in the United Kingdom and several external validation datasets, the authors provide convincing evidence that *Salmonella* genomic data can be used to identify the likely geographic source of a food-borne outbreak and, in most cases, correctly identify the country of origin of an infection acquired overseas. The work presents an excellent case for the potential utility of routine genomics coupled with machine learning for public health microbiology and the methods are likely to be applicable to other pathogens besides *Salmonella enterica*.

---

## [Decision Letter]

**Decision letter after peer review:**

Thank you for submitting your article "Rapid geographical source attribution of *Salmonella enterica* serovar Enteritidis genomes using hierarchical machine learning" for consideration by *eLife*. Your article has been reviewed by 3 peer reviewers, one of whom is a member of our Board of Reviewing Editors, and the evaluation has been overseen by Neil Ferguson as the Senior Editor. The following individuals involved in review of your submission have agreed to reveal their identity: Nicole E Wheeler (Reviewer #2); Leonid Chindelevitch (Reviewer #3).

Essential revisions:

1. It would be helpful expand on the comparison with existing methods giving more background on current state-of-the-art for geographic source attribution prior to this paper.

2. The training dataset appears to be only based on infections acquired overseas. It would be helpful to discuss the limitations in using this data source to investigate infections due to imported contaminated food.

3. More clearly communicating when a prediction is uncertain could be helpful in dealing with isolates from countries or food transport networks where it is hard to make a reliable determination. It would be helpful to consider this in the discussion.

4. There is a need for a discussion of limits to the utility of the tool due to exclusion of UK *Salmonella* isolates.

5. There is a need to improve the clarity of Figure 5, in particular increasing the resolution of the trees.

6. The authors should elaborate on the plausibility of missing data on multi-country trips and their frequency based on available travel data.

7. Any corrections made in the analysis for increased travel during the summer months should be stated.

8. Consider expanding the discussion on the poor predictions in the outbreak due to Polish eggs.

*Reviewer #1 (Recommendations for the authors):*

1. It would have been nice to see more background on state-of-the art for geographic source attribution prior to this paper. Currently there is just one sentence and two references (line 73-77).

2. Should define "SNP" on first use (line 109) also please explain "5-SNP cluster by SNP Address".

3. line 165: "was made up of 15 " or 16?

4. For a general audience (including public health professionals) it would be helpful to spend more time explaining key terminology eg "multi-class classifier", 'F1 score" etc. The work should be of wide interest, including to public health professionals, and explaining terminology would likely help increase impact of the paper.

5. line 166-167 "were predictable" What does this mean?

6. line 331 "Variation in classification accuracy was negatively associated with both low sample number and increasing within-class genetic diversity" is this the intended meaning or is what is meant "classification accuracy was negatively associated with both low sample number and increasing within-class genetic diversity" (i.e. is the association with classification accuracy or with variation in classification accuracy?).

*Reviewer #2 (Recommendations for the authors):*

Overall, this was an excellent paper to read. One thing I wish the authors had given more real estate to in the introduction is allowing the reader to understand how (if at all) the same determinations re: geographic origin are made today – how long does it take? Is there a big cost to an investigation? Are there regulatory or political hurdles?

I struggled to interpret Figure 5, and it is quite important for understanding this algorithm's capabilities. The main issue is the resolution of the trees, plus their size makes it difficult to discern how much overlap there is between the public and UKHSA data. The Microreact view makes this easier to explore and could be leveraged to create clearer figures. What I want to see in this figure is how interspersed the outbreak isolates are with UKHSA ones, which requires a close view and high resolution, and how interspersed isolates from different countries are for these outbreaks. When looking at the Poland/Spain mix-up in panel B on Microreact, it becomes very clear why this has happened when you look at the country of travel associated with each sample and the dataset each sample comes from.

Useful additional metadata columns on the Microreact tree would be the individual country of origin predictions and their associated probabilities – it would be helpful to see how confidence levels change across the tree and with more geographically interspersed clades.

Usually, I find F1 to be the best measure of accuracy, but here the % accurate calls would also be interesting, as this has direct public health implications, and there is a relationship between the total number of infections originating in a country and the accuracy of classification.

*Reviewer #3 (Recommendations for the authors):*

In addition to the concerns outlined above, here are a few typos that should be corrected; a non-exhaustive list:

– line 105: replace "from" by "to".

– line 247: providing.

– line 348: "before BEING passed ON to".

Also, just because a country has fewer than 10 samples in the collection does not mean it is "low incidence"; technically, this only means that it is "low incidence in UK tourists". Please clarify your statement accordingly.

The data and code are fully available, in compliance with *eLife*'s policies.

---

## [Author Response]

Essential revisions:1. It would be helpful expand on the comparison with existing methods giving more background on current state-of-the-art for geographic source attribution prior to this paper.

We have addressed this with an expansion of the introduction to include further sources on existing methods. The text now reads:

“The approach taken by the UKHSA for inferring geographical source from genomics data involves phylogenetic population structure analysis, the examination of the genetic relatedness of isolates from various relevant source populations, alongside in-depth analysis of isolates from recent-reported travel cases and reconstruction of domestic or international food networks (Chan et al., 2023, Chattaway et al., 2023, Jenkins et al., 2019). Due to the resources involved, this type of investigation is often only undertaken in exceptional cases representing a threat to public health. A range of other methodologies exist for application to source attribution of genomics data (Mughini-Gras et al., 2018) including; population genetics approaches such as STRUCTURE (Pritchard et al., 2000) and the asymmetric island model (Wilson et al., 2008); Bayesian methods, such as sourceR (Miller et al., 2017); and hybrid methodologies which can variably include additional information, such as exposure risk (Ravel et al., 2017). These methodologies require significant bioinformatic expertise and research time, are computationally expensive and scale poorly with the increasingly vast collections of bacterial genomes available for analysis.” Furthermore a second section was later added including background on the previous use of ML for source attribution “Machine learning (ML) approaches have recently shown utility in predicting the original host of infectious *S. enterica* and *Escherichia coli* (Lupolova et al., 2017, 2019; Wheeler et al. 2018, Zhang et al., 2019), *Campylobacter jejuni* (Arning et al. 2021), S. Typhimurium (Munk et al., 2020), and *Listeria monocytogenes* (Tanui et al., 2022), but has yet to be applied to the problem of geographical source attribution.”

2. The training dataset appears to be only based on infections acquired overseas. It would be helpful to discuss the limitations in using this data source to investigate infections due to imported contaminated food.

We have addressed this point with the addition of two separate sections of the discussion:

Firstly, addition text was added in relation to the ‘risk’ of infection per country. The text reads:

“The risk of infection per country is likely to be a product of the overall prevalence of S. Enteritidis as well as the local circulation of high-virulence or low-virulence strains, infections with which is more or less likely to result in referral to the clinical setting.”

A second section discusses the use of recently reported travel data and its limitations:

**“**Another limitations of this work was its reliance on recent reported travel data, which was used a proxy for the prevalence of infections arising from imported foodstuffs. Whilst there is evidence that this approach can capture highly-relevant disease strains present in foodstuffs imported to the UK (Chan et al., 2023, Chattaway et al., 2023, Jenkins et al., 2019), strain prevalence is likely to vary across national borders due to differences in national and domestic regulations and food-processed methods.”

3. More clearly communicating when a prediction is uncertain could be helpful in dealing with isolates from countries or food transport networks where it is hard to make a reliable determination. It would be helpful to consider this in the discussion.

We have added text, amended Figure 3 and updated the Microreact project to address this. Text was added in the ‘Granular predictions are provided at a regional, subregional and individual country level’ section when reporting on the outputs of the hML model:

“This model allows for a separate measures of predictive confidence (i.e. the probability value reported by the appropriate classifier) to be reported to the end-user for each hierarchical level. For an example of how this uncertainty maybe visualised for the end-user is available in Figure 3C and mirrored in the interactive Microreact project. These figures include three columns containing the predicted class of each sample per hierarchical level and three matched columns with the probability of the prediction per class as reported by the appropriate classifier. This also allows for a user to visually inspect the variable level of predictive confidence for each bacterial lineage within its particular phylogenetic context.”.

Additionally, the microreact figure has been recreated (https://microreact.org/project/vsy9Ctr1dbzr49Mjs53bYj-s-enteritidis-hml-classifier) and now includes the top prediction of region/subregion/country for each isolates in the training/test/validation datasets, the associated p-value and a binary measure of if the prediction was correct. The tree now defaults to including this information as coloured bars next to the tree and the colours used are congruent with those used in the main manuscript. If any of these variables are selected as tip labels to the tree then the concordance between metadata and phylogeny/geography can be observed alongside the phylogenetic tree or as piecharts on the map view.

We have added an additional panel to Figure 3 (Figure 3C) which contains the same data presented in the Microreact project described above with matching colours for consistency of presentation.

4. There is a need for a discussion of limits to the utility of the tool due to exclusion of UK Salmonella isolates.

We have added text to address this point in the discussion of future directions for this work. The additional text reads:

“Furthermore, the UK was excluded as a potential geographical source in the model due to extremely low prevalence of *S.* Enteritidis in domestic poultry flocks and a lack of clinical isolates linked to consumption of locally produced foodstuffs (European Food Safety Authority and European Centre for Disease Prevention and Control, 2017). Future studies could include samples collected via veterinary or food chain-based surveillance programmes to provide finer scale inference of causation or provide model coverage of low-prevalence geographical locations.”

5. There is a need to improve the clarity of Figure 5, in particular increasing the resolution of the trees.

To improve clarity of Figure 5 we have condensed the multiple phylogenic trees on the right hand side of the figure to display a single phylogenetic tree containing all of the training and validation samples. Coloured tree tips representing the training data from countries matching those in the validation dataset. The samples from the validation dataset along with the associated p-value of predictions for region/subregion/country have been displayed as additional bars alongside the tree. The figure legend has been updated to reflect this change. A supplementary figure (Figure 5 —figure supplement 1) has been added which shows the same data but in a circularised form which maybe easier for some readers to interpret.

6. The authors should elaborate on the plausibility of missing data on multi-country trips and their frequency based on available travel data.

We believe that this is a good point for discussion, but one which is currently difficult to investigate due to the limitations of the available travel data. We have added a section to the discussion discussion these limitations and speculating where we are able. The text reads:

“Other potential forms of mislabelling in reported travel data includes the incorrect reporting for multi-country trips or infection during layovers during return travel. The ONS records travel destination as the primary location visited, so no estimates of the proportion of samples attributable to multi-country trips could be made, but will certainly represent a minority of the data. Recent surveys have estimated that layovers occur on only a small proportion of flights (~15%), although this could potentially have a larger impact on the data due to the bias in detecting cases which occur towards the end of a trip, which could result in a patient being admitted to UK-based clinical care setting, rather than one overseas (Department for Transport, 2018a, Department for Transport, 2018b).“

7. Any corrections made in the analysis for increased travel during the summer months should be stated.

No correction for increased travel during the summer months was made in this analysis. In order to clarify this, the text in the discussion was changed from “The model presented herein is conceptually simple and does not incorporate temporal information, … ” changed to “ The model presented herein is conceptually simple and does not incorporate temporal information, account for seasonal fluctuations in disease incidence,…”. Furthermore, the figure legend for Figure 1 has the following text added:

“No correction was made for the seasonal variation observed in international travel.”

8. Consider expanding the discussion on the poor predictions in the outbreak due to Polish eggs.

Our thanks to the reviewers for identifying this useful improvement to the manuscript. An audit was taken of training samples which had likely originated from the polish outbreak present in the training dataset. This identified only 6 samples from 4 countries. None of these were attributed to a Polish origin, but 5/6 samples came from Southern Europe. The majority of the Polish eggs outbreak samples were misattributed to a Southern European origin and this analysis providing a more parsimonious explanation than we have previously provided. Text has been added in the Results section to reflect this:

“It was possible to identify samples attributed to the outbreak in our training data, this included 6 samples which were present in the training dataset. These six samples were labelled as having been acquired from Portugal (3), Spain (1), Greece (1) and Turkey (1) based on recent reported travel information, despite having got their infection from Polish eggs. 5/6 of these were of a Southern European origin, and likely contributed to the misattribution of the samples from the Polish validation datasets to a Southern European origin..”

The summary text at the bottom of this section was amended to read:

“A large-scale, long-term, multi-country outbreak was poorly predicted by the model, due to a small number of outbreak isolates being present in the training dataset but incorrectly labelled as having a Southern European origin”.

Further small changes have been made throughout the manuscript related to this addition.

Reviewer #1 (Recommendations for the authors):1. It would have been nice to see more background on state-of-the art for geographic source attribution prior to this paper. Currently there is just one sentence and two references (line 73-77).

Please refer to the additions made during Essential Revision 1.

2. Should define "SNP" on first use (line 109) also please explain "5-SNP cluster by SNP Address".

Text at line 109 was chaged to read:

“Clones were defined as a single-linkage cluster of isolates with 5 or less single nucleotide polymorphisms (SNPs), a definition routinely-used by the UKHSA in genomic disease surveillance (Chattaway et al., 2019, Dallman et al., 2018). In order to reduce the influence of highly related clonal outbreaks on the resulting ML model, a single, random representative isolate was selected per clone per country.”

3. line 165: "was made up of 15 " or 16?

Amended in text to “16”.

4. For a general audience (including public health professionals) it would be helpful to spend more time explaining key terminology eg "multi-class classifier", 'F1 score" etc. The work should be of wide interest, including to public health professionals, and explaining terminology would likely help increase impact of the paper.

We thank the reviewer for this advice. To this end we added these explanations in text related to multiple topics:

a) population structure analysis – “population structure analysis, the examination of the genetic relatedness of isolates from various relevant source populations.”

b) Multi-class classifiers – “Multi-class classifiers allow for the classification of samples into 3 or more classes (i.e. Region = Europe, Africa, Americas, Asia), rather than making a series of binary classifiers per individual class (e.g. Africa/Not-Africa, Europe/Not-Europe etc.).”

c) F1-score – “Model were assessed using a range of accuracy metrics, although the F1-score (or its hierarchical alternative, hF1) was preferentially used. The F1-score or F-score is the harmonic mean between the precision (how much samples are correctly classified as positive out of all positives) and recall (how many positive samples are predicted as positive) and allows for a partial representation of both values in a single metric.”

d) Resampling data- “Due to the imbalanced nature of the real-world surveillance dataset, it was necessary resample the data to generate less biased datasets for training.”

5. line 166-167 "were predictable" What does this mean?

Clarified in text:

“In total, 53 individual classes (4 regions, 11 sub-regions and 38 countries) were included in the model.”

6. line 331 "Variation in classification accuracy was negatively associated with both low sample number and increasing within-class genetic diversity" is this the intended meaning or is what is meant "classification accuracy was negatively associated with both low sample number and increasing within-class genetic diversity" (i.e. is the association with classification accuracy or with variation in classification accuracy?).

Line 331 now reads:

“Classification accuracy was negatively associated with both low sample number and increasing within-class genetic diversity”

Reviewer #2 (Recommendations for the authors):Overall, this was an excellent paper to read. One thing I wish the authors had given more real estate to in the introduction is allowing the reader to understand how (if at all) the same determinations re: geographic origin are made today – how long does it take? Is there a big cost to an investigation? Are there regulatory or political hurdles?

The references added to address Essential Revision 1 represent the current state-of-the-art methods used for genomic investigation of outbreaks by the UKHSA.

I struggled to interpret Figure 5, and it is quite important for understanding this algorithm's capabilities. The main issue is the resolution of the trees, plus their size makes it difficult to discern how much overlap there is between the public and UKHSA data. The Microreact view makes this easier to explore and could be leveraged to create clearer figures. What I want to see in this figure is how interspersed the outbreak isolates are with UKHSA ones, which requires a close view and high resolution, and how interspersed isolates from different countries are for these outbreaks. When looking at the Poland/Spain mix-up in panel B on Microreact, it becomes very clear why this has happened when you look at the country of travel associated with each sample and the dataset each sample comes from.

Figure 5 has been amended for clarity and the Microreact project has been recreated with additional metadata and the outputs of the model to allow interactive visualisation of model outputs and uncertainty (https://microreact.org/project/vsy9Ctr1dbzr49Mjs53bYj-s-enteritidis-hml-classifier). An additional supplementary figure (Figure 5) has been added to aid visualisation.

Please see Essential Revision 5 for more details.

Useful additional metadata columns on the Microreact tree would be the individual country of origin predictions and their associated probabilities – it would be helpful to see how confidence levels change across the tree and with more geographically interspersed clades.

This information has been included in the new Microreact project (please see Essential Revision 5 or the previous comment).

Usually, I find F1 to be the best measure of accuracy, but here the % accurate calls would also be interesting, as this has direct public health implications, and there is a relationship between the total number of infections originating in a country and the accuracy of classification.

We have added an additional supplementary figure (Supp. Figure *) which is a modification of Figure 3 which includes %correctly predicted per class. Text has been added to the results to reflect this:

“For ease of interpretation, an alternative to Figure 3 has been including in the supplementary information (Figure 3 —figure supplement 3) which simply displays the % of isolates of each class correctly assigned to that class instead of HF1.”

Reviewer #3 (Recommendations for the authors):In addition to the concerns outlined above, here are a few typos that should be corrected; a non-exhaustive list:– line 105: replace "from" by "to".

Replaced in text

– line 247: providing.

Replaced in text

– line 348: "before BEING passed ON to".

Replaced in text

Also, just because a country has fewer than 10 samples in the collection does not mean it is "low incidence"; technically, this only means that it is "low incidence in UK tourists". Please clarify your statement accordingly.

Clarified in text:

“After quality-filtering, downsampling and the removal of countries with a low-incidence of reported infections of UK tourists (<10 total isolates)”